# MAXIMUM $n$-TIMES COVERAGE FOR VACCINE DESIGN

**Ge Liu, Alexander Dimitrakakis, Brandon Carter, David Gifford**
Computer Science and Artificial Intelligence Laboratory
Massachusetts Institute of Technology
gifford@mit.edu

## ABSTRACT

We introduce the maximum $n$-times coverage problem that selects $k$ overlays to maximize the summed coverage of weighted elements, where each element must be covered at least $n$ times. We also define the min-cost $n$-times coverage problem where the objective is to select the minimum set of overlays such that the sum of the weights of elements that are covered at least $n$ times is at least $\tau$. Maximum $n$-times coverage is a generalization of the multi-set multi-cover problem, is NP-complete, and is not submodular. We introduce two new practical solutions for $n$-times coverage based on integer linear programming and sequential greedy optimization. We show that maximum $n$-times coverage is a natural way to frame peptide vaccine design, and find that it produces a pan-strain COVID-19 vaccine design that is superior to 29 other published designs in predicted population coverage and the expected number of peptides displayed by each individual's HLA molecules.

## 1 INTRODUCTION

In the maximum $n$-times coverage problem, a set of overlays is selected to cover elements zero or more times, where each overlay is predetermined to cover one or more elements. Each element is assigned a weight that reflects its importance. The objective of the problem is to maximize the sum of the weights of elements that are covered at least $n$ times by at most $k$ overlays. When the coverage of elements by overlays is determined by a machine learning method, the result summarizes machine learning results with a compact solution that is shaped by the element weights employed.

Our introduction of weighted elements and required $n$-times coverage creates a new class of problem without a known solution or complexity bound for approximate solutions. The closest problem to the maximum $n$-times coverage problem is the *multi-set multi-cover problem* which does not assign weights to the elements and thus can be formulated as the Covering Integer Program (CIP) problem (Srinivasan, 1999). Thus like CIP, maximum $n$-times coverage is NP-complete. A $\log(n)$-time approximation algorithm for CIP can violate coverage constraints (Dobson, 1982; Kolliopoulos, 2003; Kolliopoulos & Young, 2001). Deletion-robust submodular maximization protects against adversarial deletion (Bogunovic et al., 2018; Mirzasoleiman et al., 2017) and robust submodular optimization protects against objective function uncertainty (Iyer, 2019), but neither guarantees that each element is covered $n$ times.

In this work, we investigate properties of the maximum $n$-times coverage problem, provide a practical solution to the problem, and use the solution for machine learning based vaccine design. We show that maximum $n$-times coverage is not submodular, and introduce NTIMES-ILP and MARGINALGREEDY, efficient algorithms for solving the $n$-times coverage problem on both synthetic data and real vaccine design. In our framing of vaccine design, an element is a specific collection of HLA alleles (a genotype), weights are the frequency of genotypes in the population, $n$ is the desired number of peptides displayed by each individual, and an overlay is a peptide that is predicted to be displayed by each genotype a specified number of times. The solution of the maximum $n$-times coverage problem allows us to find a set of overlays that maximizes the sum of element weights (population coverage). We show that framing vaccine design as maximum $n$-times coverage produces a solution that produces superior predicted population coverage when compared to 29 previous published vaccines for COVID-19 with less than 150 peptides.

## 2 THE MAXIMUM $n$-TIMES COVERAGE PROBLEM

### 2.1 MULTI-SET MULTI-COVER PROBLEM

In the standard SET COVER and MAXIMUM COVERAGE problems, we are given a set $\mathcal{U}$ of $|\mathcal{U}|$ elements (also known as the universe) and a collection $\mathcal{S} = \{S_1, S_2, \ldots, S_m\}$ of $m$ subsets of $\mathcal{U}$ such that $\bigcup_i S_i = \mathcal{U}$. The goal in the SET COVER problem is to select a minimal-cardinality set of subsets from $\mathcal{S}$ such that their union covers $\mathcal{U}$. The MULTI-SET MULTI-COVER (MSMC) problem is a generalization of the SET COVER problem, where multi-sets are sets in which an element can appear more than once. The objective of the MSMC problem is to determine the minimum number of multisets (a multi-set can be chosen multiple times) such that each element $i$ is covered at least $b_i$ times. It can be formulated into Covering Integer Program (CIP) problem (Srinivasan, 1999):

**Definition 1.** *(Covering Integer Program, CIP) Given* $A \in \mathbb{R}_+^{n \times m}, b \in \mathbb{R}_+^n, w \in \mathbb{R}_+^m, d \in \mathbb{R}_+^m$, *a CIP* $P = (A, b, w, d)$ *seeks to minimize* $w^T x$, *subject to* $Ax \geq b, x \in \mathbb{Z}_+^m$, *and* $x \leq d$.

Here $A_{ij}$ represents the number of times $i$-th element appears in the $j$-th multi-set. The $w$ is set to be all 1 in MSMC. The constraints $x \leq b$ are called multiplicity constraints which limit the number of times a single multi-set can be reused, and they generally make covering problems much harder as natural linear programming (LP) relaxation can have an unbounded integrality gap (Chuzhoy & Naor, 2006). Dobson (1982) provides a combinatorial greedy $H(\max_j \sum_i A_{ij})$-approximation algorithm ($H(t)$ stands for $t$-th harmonic number) but multiplicity constraints can be dealt with effectively only in the (0,1) case, and thus this algorithm can be as bad as polynomial. Kolliopoulos & Young (2001); Kolliopoulos (2003) gave a tighter-bound solution that can obtain $O(\log n)$-approximation.

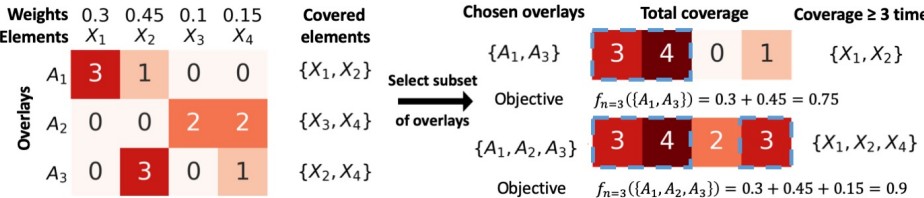

Figure 1: Example of $n$-times coverage calculation.

### 2.2 MAXIMUM $n$-TIMES COVERAGE

We introduce the maximum $n$-times coverage problem, a variant of the MSMC problem that accounts for multiple coverage of each element while also assigning weights to different elements. We are given a set $\mathcal{X}$ with $l$ elements $\{X_1, X_2, \ldots, X_l\}$ each associated with a non-negative weight $w(X_i)$, and a set of $m$ overlays $\mathcal{A} = \{A_1, A_2, \ldots, A_m\}$. Each overlay $A_j$ covers each element $i$ in $\mathcal{X}$ an element specific number of times $c_j(X_i)$, which is similar to a multi-set. When $c_j(X_i) = 0$ the element $X_i$ is not covered by overlay $A_j$. We use a very strict multiplicity constraint such that each overlay can be used only once. Given a subset of overlays $O \subseteq \mathcal{A}$, the total number of times an element $X_i$ is covered by $O$ is the sum of $c_j(X_i)$ for each overlay $j$ in $O$:

$$C(X_i|O) = \sum_{j \in O} c_j(X_i) \tag{1}$$

We define the *n-times coverage function* $f_n(O)$ as the sum of weights of elements in $\mathcal{X}$ that are covered at least $n$ times by $O$. Figure 1 shows an example computation of the *n-times coverage function*.

$$f_n(O) = \sum_{i=1}^{l} w(X_i) \mathbb{1}_{\{C(X_i|O) \geq n\}} = \sum_{i=1}^{l} w(X_i) \mathbb{1}_{\{\sum_{j \in O} c_j(X_i) \geq n\}} \tag{2}$$

The objective of the MAXIMUM $n$-TIMES COVERAGE problem is to select a set of $k$ overlays $O \subseteq \mathcal{A}$ such that $f_n(O)$ is maximized. This can be formulated as the maximization of the monotone set function $f_n(O)$ under cardinality constraint $k$:

$$O^* = \underset{O \subseteq \mathcal{A}, |O| \leq k}{\arg\max} f_n(O) \tag{3}$$

We define MIN-COST $n$-TIMES COVERAGE as the minimum set of overlays such that the sum of the weights of elements covered at least $n$ times is $\geq \tau$. We assume $\mathcal{A}$ provides sufficient $n$-times coverage for $f_n(\mathcal{A}) \geq \tau$. We define the $n$-TIMES SET COVER problem as the special case of MIN-COST $n$-TIMES COVERAGE where $\tau = \sum_i w(X_i)$.

$$\text{MIN-COST } n\text{-TIMES COVERAGE} \qquad O^* = \arg\min_O |O| \quad \text{s.t.} \quad f_n(O) \geq \tau \tag{4}$$

$$n\text{-TIMES SET COVER} \qquad O^* = \arg\min_O |O| \quad \text{s.t.} \quad f_n(O) = f_n(\mathcal{A}) \tag{5}$$

**Theorem 1.** *The $n$-times coverage function $f_n(O)$ is not submodular.*

The proof can be found in Appendix A. Since the $n$-times coverage problem is not submodular, we cannot take advantage of the proven near-optimal performance of the greedy algorithm for the SUBMODULAR MAXIMUM COVERAGE problem. Thus we seek new solutions.

## 3   THE NTIMES-ILP SOLUTION FOR MAXIMUM $n$-TIMES COVERAGE

We first formulate the maximum-$n$ times coverage problem as an integer linear program (ILP). We associate a binary variable $a_i$ with each overlay $A_i$ such that $a_i = 1$ if $A_i \in O^*$ and otherwise $a_i = 0$. Thus the cardinality constraint can be written as $|O| = \sum_{i=1}^m a_i \leq k$, and the number of times an element $X_j$ is covered by $O$ is $C(X_j|O) = \sum_{i \in O} c_i(X_j) = \sum_{i=1}^m a_i c_i(X_j)$. The main challenge is to encode the objective function $f_n(O) = \sum_{j=1}^l w(X_j) \mathbb{1}_{\{\sum_{i=1}^m a_i c_i(X_j) \geq n\}}$ in a linear fashion. We replace the step function $\mathbb{1}_{\{\sum_{i=1}^m a_i c_i(X_j) \geq n\}}$ with a variable $t_j$ such that for each element $X_j$:

$$t_j = 1 \iff \sum_{i=1}^m a_i c_i(X_j) \geq n \quad \text{and} \quad t_j = 0 \iff \sum_{i=1}^m a_i c_i(X_j) < n \tag{6}$$

We enforce the conditions in (6) with the Big M method (Sherali et al., 2013), where we include the following inequalities for each element $X_j \in \mathcal{X}$:

$$-M(1 - t_j) \leq \sum_{i=1}^m a_i c_i(X_j) - n + \epsilon \leq M t_j, \tag{7}$$

We set $\epsilon \in (0, 1)$ and choose $M$ to be a large number such as 10,000 which is larger than the term $\sum_{i=1}^m a_i c_i(X_j) - n + \epsilon$. Here we prove that $t_j = 1 \iff \sum_{i=1}^m a_i c_i(X_j) \geq n$.

*Proof.* When $t_j = 1$, inequality (7) becomes $0 \leq \sum_{i=1}^m a_i c_i(X_j) - n + \epsilon \leq M$. If $M$ is a large number, we can ignore the right inequality and rearrange to get $\sum_{i=1}^m a_i c_i(X_j) \geq n - \epsilon$. Since $\sum_{i=1}^m a_i c_i(X_j)$ and $n$ are both integers and $\epsilon \in (0, 1)$, we find $\sum_{i=1}^m a_i c_i(X_j) \geq n$. All these steps can be taken in the reverse order to prove that $\sum_{i=1}^m a_i c_i(X_j) \geq n \implies t_j = 1$, given that $t_j$ is a binary variable. $\qquad \square$

We also prove that $t_j = 0 \iff \sum_{i=1}^m a_i c_i(X_j) < n$.

*Proof.* We start by showing the forward direction. If $t_j = 0$, then inequality (7) becomes $-M \leq \sum_{i=1}^m a_i c_i(X_j) - n + \epsilon \leq 0$. As $M$ is large, $-M$ is a large negative number, so we can ignore the left inequality. Hence, $\sum_{i=1}^m a_i c_i(X_j) \leq n - \epsilon < n$. In the reverse direction, if $\sum_{i=1}^m a_i c_i(X_j) < n$, because both the quantities $\sum_{i=1}^m a_i c_i(X_j)$ and $n$ are integers, for any $\epsilon \in (0, 1)$, $\sum_{i=1}^m a_i c_i(X_j) \leq n - \epsilon$. Rearranging this we get $\sum_{i=1}^m a_i c_i(X_j) - n + \epsilon \leq 0$, which forces $t_j$ as a binary variable to be equal to 0 from inequality (7). $\qquad \square$

Therefore, the objective function to maximize becomes $f_n(O) = \sum_{j=1}^l w(X_j) \mathbb{1}_{\{C(X_j|O) \geq n\}} = \sum_{j=1}^l t_j w(X_j)$, which is linear as a sum over $l$ terms of a binary variable multiplied by a constant.

The complete NTIMES-ILP formulation of $n$-times coverage is:

- Variables

- – $a_1, ..., a_m$ representing presence of each overlay $A_1, ..., A_m$ in final solution
  - – $t_1, ..., t_l$ representing if each element $X_1, ..., X_l$ has been covered at least $n$ times
- • Constraints
  - – $\sum_{i=1}^{m} a_i \leq k$ (the maximum total number of overlays allowed in the final subset)
  - – $-M(1 - t_j) \leq \sum_{i=1}^{m} a_i c_i(X_j) - n + \epsilon \leq M t_j$ for each $j \in \{1, ..., l\}$
- • Objective to maximize
  - – $f_n(O) = \sum_{j=1}^{l} w(X_j) \mathbb{1}_{\{C(X_j|O) \geq n\}} = \sum_{j=1}^{l} t_j w(X_j)$

If we want to enforce non-redundancy constraints such that pairs of overlays that violate certain distance criteria are not both chosen, we can include additional constraints $a_t + a_r \leq 1$ for every pair of overlays $(A_t, A_r)$ that we don't want both to be included in the final subset.

## 4 THE MARGINALGREEDY ALGORITHM FOR MAXIMUM $n$-TIMES COVERAGE

Although NTIMES-ILP can produce near-optimal solutions on problems with reasonable size, it may become intractable for problems where hundreds of thousands of elements and/or overlays are involved, such as certain variants of the vaccine design problem. Thus, we seek a polynomial time algorithm that provides good solutions to the maximum $n$-times coverage problem. A naive greedy solution is problematic when the $n$-times objective is directly approached. This is a consequence of potential early bad overlay choices by a greedy approach that can cause it to fail later to find overlays with sufficient $n$-times coverage. In addition, during greedy optimization available overlay choices may not provide differential marginal gain to avoid ties and random overlay selection. The MARGINALGREEDY algorithm is specifically designed to avoid early bad choices that will lead to failure by marginally approaching the $n$-times coverage objective. MARGINALGREEDY preserves marginal gains by employing look-ahead tie-breaking that assists in selecting overlays that benefit longer term objectives.

---

**Algorithm 1** MARGINALGREEDY algorithm (for MIN-COST $n$-TIMES COVERAGE)

---

**Input:** Weights of the elements in $\mathcal{X}$: $w(X_1), w(X_2), \ldots, w(X_l)$, ground set of overlays $\mathcal{A}$ where each overlay $j$ in $\mathcal{A}$ covers $X_i$ for $c_j(X_i)$ times, target minimum # times being covered $n_{target}$, coverage cutoff for different $n$: $\tau_1, \tau_2, \ldots, \tau_{n_{target}}$, beam size $b$

Initialize beam $B^0 \leftarrow \{\emptyset\}$, $t = 0$

**for** $n = 1, \ldots, n_{target}$ **do**

    Using set function $f_n(S) = \sum_{i=1}^{l} w(X_i) \mathbb{1}_{\{\sum_{j \in S} c_j(X_i) \geq n\}}$ as objective function.

    **repeat**

        $K^t \leftarrow \emptyset$                                  $\triangleright$ $K^t$ is the set of candidate solutions

        **for** $O \in B^t$ **do**                              $\triangleright$ Beam search

            **for** $a \in \mathcal{A} \setminus O$ **do**

                $K^t \leftarrow K^t \cup \{O \cup \{a\}\}$         $\triangleright$ Add the candidate solution $\{O \cup \{a\}\}$ to $K^t$

        $B^{t+1} \leftarrow \{\text{Top } b \text{ candidate solutions in } K^t \text{ as scored by } f_n(S)\}$

        $t \leftarrow t + 1$

        $m^t \leftarrow$ median score of candidate solutions in the beam $B^t$

    **until** $m^t \geq \tau_n$ (in $n_{target}$-th cycle the termination condition is $\max_{S \in B^t} f_n(S) \geq \tau_{target}$)

    (for MAXIMUM $n$-TIMES COVERAGE with cardinality constraint $k$, there is additional termination condition $t > k$)

    $O^* = \arg\max_{S \in B^t} f_{n_{target}}(S)$

**Output:** The final selected subset of overlays $O^*$

---

MARGINALGREEDY optimizes $n$-times coverage with a sequence of greedy optimization cycles where the $n$-th cycle optimizes the coverage function $f_n(S)$. We establish coverage starting at $n = 1$ and incrementally increase the $n$-times criteria to $n_{target}$. Thus early overlay selections are guided by less stringent cycle-specific coverage objectives. A set of coverage cutoffs $\{\tau_1, \tau_2, \ldots, \tau_{target}\}$ is used as the termination condition for each greedy optimization cycle, and when not specified, we assume $\tau_1 = \tau_2 = \cdots = \tau_{target}$ by default. We use beam search to keep track of top $b$ candidate

solutions at each iteration. In general the larger the beam size, the closer the result is to the true optimal. However, there is a tradeoff between beam size and running time. In our results we choose the largest beam size that maintains a practical running time as we describe below.

The full algorithm is given in Algorithm 1. A similar algorithm can be used to solve the $n$-TIMES SET COVER problem in which $\tau_{target} = f_{n_{target}}(\mathcal{A})$. For the MAXIMUM $n$-TIMES COVERAGE problem with cardinality constraint $k$, the optimization terminates when $t > k$. When beam search is not used to reduce computation time ($b = 1$), we have extended the algorithm to break ties during the $n$-times coverage iteration by looking ahead to $(n + 1)$-times coverage. We call this extension *look ahead tie-breaking*. Another advantage of MARGINALGREEDY is that it is capable of guaranteeing high coverage for $n_t < n_{target}$ by controlling the cycle-specific coverage cutoffs $\tau_t$. This is desired in vaccine design where wider population coverage is also important and we want to make sure that almost 100% of the population will be covered at least once.

## 5 VACCINE POPULATION COVERAGE MAXIMIZATION

Contemporary peptide vaccine design methods use machine learning scoring of peptide display for HLA alleles followed by the selection of peptides to maximize 1-times coverage. These methods do not accurately model the frequency of HLA haplotypes in a population and thus can not accurately assess the population coverage provided by a vaccine. Malone et al. (2020) uses haplotypes but does not explicitly model their frequencies. For a selected set of peptides, the IEDB Population Coverage Tool (Bui et al., 2006) estimates peptide-MHC binding coverage and the distribution of peptides displayed for a given population but does not consider linkage disequilibrium between HLA loci.

We frame vaccine design as maximum $n$-times coverage because ideally each vaccinated individual in a population will be "covered" by multiple immunogenic peptides. While it might be assumed that an individual will be vaccinated if they display a single peptide, three independent lines of reasoning support the need for $n$-times coverage:

1. When an individual displays multiple peptides their immune system activates and expands more than one set of T cell clonotypes that are poised to fight viral infection (Sekine et al., 2020; Schultheiß et al., 2020; Grifoni et al., 2020).

2. The peptides that are immunogenic vary from one individual to another, and thus having multiple peptides displayed increases the probability at least one will be strongly immunogenic (Croft et al., 2019).

3. If a virus evolves and changes its peptide composition, using multiple peptides reduces the chance of viral escape (Wibmer et al., 2021).

Prior work has not considered formalizing these aspects of vaccine design with an $n$-times constraint and thus has produced solutions to the 1-times coverage task. Existing solutions to 1-times coverage do not anticipate or solve the $n$-times coverage task. Both Malone et al. (2020) and Toussaint et al. (2008) provide solutions to 1-times coverage, Lundegaard et al. (2010) does not provide specific population coverage guarantees, and Oyarzun & Kobe (2015) reviews methods for 1-times coverage. Discrete optimization has been used for other aspects of vaccine design that are unrelated to population coverage, such designing a single peptide sequence that covers a set of diverse but related set of input epitopes (Theiler & Korber, 2018), and designing spacers for string-of-beads peptide delivery (Schubert & Kohlbacher, 2016).

For a peptide to be effective in a vaccine it must be presented by an individual's Major Histocompatibility Complex (MHC) molecules and be immunogenic. A displayed peptide is immunogenic when it activates T cells, which expand in number and mount a response against pathogens or tumor cells. Memory T cells provide robust immunity against COVID-19, even in the absence of detectable antibodies (Sekine et al., 2020). The use of peptide vaccine components to activate T cells is in development for cancer (Hu et al., 2018) and viral diseases including HIV (Arunachalam et al., 2020), HPV (Kenter et al., 2009) and malaria (Li et al., 2014; Rosendahl Huber et al., 2014).

A challenge for the design of peptide vaccines is the diversity of human MHC alleles that each have specific preferences for the peptide sequences they will display. The Human Leukocyte Antigen (HLA) loci, located within the MHC, encode the HLA class I and class II molecules. We consider

the three classical class I loci (HLA-A, HLA-B, and HLA-C) and three loci that encode class II molecules (HLA-DR, HLA-DQ, and HLA-DP). A single chromosome contains one allele at each locus. We use *haplotype* to represent a combination of HLA alleles in a single chromosome, denoted as $A_i B_j C_k$ for MHC class I or $DR_i DQ_j DP_k$ for MHC class II. Each haplotype has a frequency $G(i, j, k)$ in a given population where $\sum_{i=1}^{|\mathcal{A}|} \sum_{j=1}^{|\mathcal{B}|} \sum_{k=1}^{|\mathcal{C}|} G(i, j, k) = 1$. Each individual inherits two haplotypes to form their diploid *genotype* and may carry 3–6 different alleles per MHC class depending on the zygosity. The frequency of a diploid genotype is thus (MHC class I as an example):

$$F^{i_1 j_1 k_1 i_2 j_2 k_2} = F(A_{i_1} B_{j_1} C_{k_1}, A_{i_2} B_{j_2} C_{k_2}) = G(i_1, j_1, k_1) G(i_2, j_2, k_2) \tag{8}$$

We use observed frequencies of 2,138 MHC class I and 1,711 MHC class II haplotypes for our population coverage optimization. Each haplotype represents a specific combination of 230 different HLA-A, HLA-B, and HLA-C alleles (MHC class I) or 280 different HLA-DP, HLA-DQ, and HLA-DR alleles (MHC class II) (Liu et al., 2020). Our use of haplotypes models the linkage disequilibrium between HLA genes. We separately select MHC class I and class II epitopes using $n$-times coverage and then combine them in a single vaccine delivery construct for an effective immune response.

We adopt an experimentally calibrated model of peptide-HLA immunogenicity to design vaccines using $n$-times coverage (Liu et al., 2021) (Appendix B). We utilize the observed immunogenicity of peptides in convalescent COVID-19 patient peripheral blood mononuclear cell samples (Snyder et al., 2020; Klinger et al., 2015; Nolan et al., 2020), and use these data to select machine learning models to predict immunogenicity for peptides or HLA alleles that are not observed in the clinical data. We define a *peptide-HLA hit* as a (peptide, HLA allele) pair where the peptide is predicted to be immunogenic and displayed by the HLA allele. Once we have determined a candidate set of peptides that are predicted to be immunogenic, we then need to select a minimal subset of peptides such that each individual in $\tau_{target}$ of the population is predicted to have at least $n$ peptide-HLA hits.

We first introduce *EvalVax-Robust*, an evaluation tool for estimating the population coverage of a proposed peptide vaccine set. *EvalVax-Robust* evaluates the percentage of the population having at least $n$ peptide-HLA binding hits in each individual. For a given peptide $p$ and a class I HLA allele $X \in \mathcal{A} \cup \mathcal{B} \cup \mathcal{C}$, our machine learning model outputs a binary hit prediction $e_p(X) \in \{0, 1\}$ indicating peptide-HLA immunogenicity. For each diploid genotype we compute the total number of peptide-HLA hits as the sum of $e_p(X)$ over the unique HLA alleles in the genotype.

$$c_p^{i_1 j_1 k_1 i_2 j_2 k_2} = c_p(A_{i_1} B_{j_1} C_{k_1}, A_{i_2} B_{j_2} C_{k_2}) = \sum_{X \in \{A_{i_1}, B_{j_1}, C_{k_1}\} \cup \{A_{i_2}, B_{j_2}, C_{k_2}\}} e_p(X) \tag{9}$$

The total number of peptide-HLA hits provided by a set of peptides $O$ is the sum of number of hits per peptide:

$$C_O^{i_1 j_1 k_1 i_2 j_2 k_2} = C(A_{i_1} B_{j_1} C_{k_1}, A_{i_2} B_{j_2} C_{k_2} | O) = \sum_{p \in O} c_p(A_{i_1} B_{j_1} C_{k_1}, A_{i_2} B_{j_2} C_{k_2}) \tag{10}$$

We define the *EvalVax-Robust* predicted population coverage with $\geq n$ peptide-HLA hits for a peptide vaccine set $P$ as:

$$f_n(O) = \sum_{i_1=1}^{|\mathcal{A}|} \sum_{j_1=1}^{|\mathcal{B}|} \sum_{k_1=1}^{|\mathcal{C}|} \sum_{i_2=1}^{|\mathcal{A}|} \sum_{j_2=1}^{|\mathcal{B}|} \sum_{k_2=1}^{|\mathcal{C}|} F^{i_1 j_1 k_1 i_2 j_2 k_2} \cdot \mathbb{1}\{C_O^{i_1 j_1 k_1 i_2 j_2 k_2} \geq n\} \tag{11}$$

EvalVax-Robust population coverage optimization can be accomplished by a solution to the maximum $n$-times coverage problem, as we can rewrite equation (11) into equation (2) by setting $\mathcal{X}$ to be the set of possible genotypes $i_1 j_1 k_1 i_2 j_2 k_2$, and the weights $w(X_i)$ to be the genotype frequencies $F^{i_1 j_1 k_1 i_2 j_2 k_2}$. The peptide candidate set is the set of possible overlays $\mathcal{A}$, where each peptide $p \in \mathcal{A}$ is an overlay which covers a genotype $c_p^{i_1 j_1 k_1 i_2 j_2 k_2}$ times. We directly applied MARGINALGREEDY on EvalVax-Robust objective function with a beam size of 10 for MHC class I and 5 for MHC class II, but we further reduced peptide redundancy by eliminating unselected peptides that are within three (MHC class I) or five (MHC class II) edits on a sequence distance metric from the selected peptides at each iteration. The same non-redundancy constraints are added to NTIMES-ILP as described in Section 3. As the number of unique genotypes is on the scale of millions (1M for MHC class I and 0.6M for MHC class II), for NTIMES-ILP we used an alternative objective $f_n(O)_{hap}$ that computes $n$-times coverage at the haplotype level for better tractability.

$$f_n(O)_{hap} = \sum_{i=1}^{|\mathcal{A}|} \sum_{j=1}^{|\mathcal{B}|} \sum_{k=1}^{|\mathcal{C}|} G(i, j, k) \cdot \mathbb{1}\{C_O^{ijk} \geq n\} \tag{12}$$

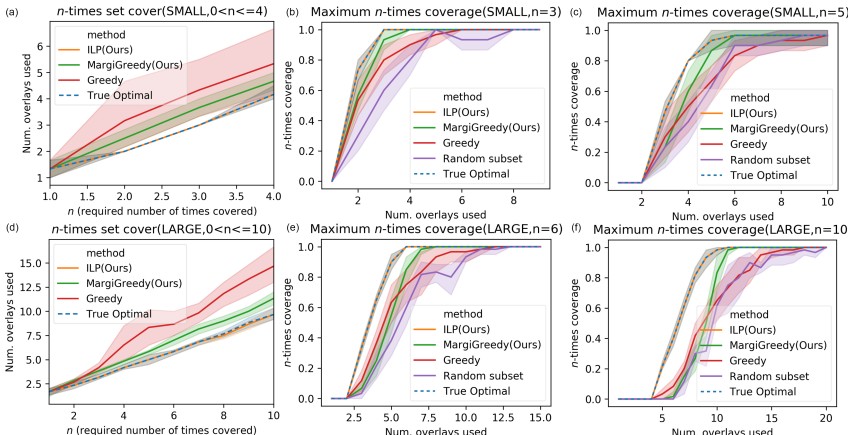

Figure 2: The MARGINALGREEDY and NTIMES-ILP algorithms outperform the greedy algorithm on both LARGE and SMALL toy examples. Superior performance is seen on both the $n$-TIMES SET COVER where a smaller number of overlays is used by MARGINALGREEDY and NTIMES-ILP to achieve 100% coverage at different criteria of $n$, and on the MAXIMUM $n$-TIMES COVERAGE problems where MARGINALGREEDY and NTIMES-ILP achieve higher coverage given same number of overlays. Shaded regions indicate 95% confidence intervals.

# 6 RESULTS

## 6.1 TOY EXAMPLES

We empirically evaluate the NTIMES-ILP and MARGINALGREEDY algorithms with two toy examples: a LARGE dataset where a set of 30 overlays are randomly generated to cover a set of 10 elements (with equal weights) 0, 1, or 2 times, and a SMALL dataset where a set of 10 overlays that were randomly generated to cover a set of 5 elements with equal weights 0, 1, or 2 times. Figure 2 shows the efficiency of the NTIMES-ILP and MARGINALGREEDY algorithms on both the $n$-TIMES SET COVER and MAXIMUM $n$-TIMES COVERAGE problems for varying values of $n$. We compare our algorithms to a greedy algorithm that directly optimizes $n$-times coverage and find the greedy algorithm degenerates to sub-optimal solutions for large values of $n$. These sub-optimal solutions can be as bad as random. We also compute the true optimal solution with exponential-time exhaustive search. We repeated problem generation and optimization with 6 different random seeds to provide confidence bounds and used beam size $b = 1$ with look-ahead tie-breaking for MARGINALGREEDY. As shown in Figure 2, NTIMES-ILP achieves true optimal in all datasets and MARGINALGREEDY outperforms greedy on all tasks and datasets. We observed that MARGINALGREEDY has a significant advantage over greedy in tests with larger $n$ and in regions of higher coverage. Appendix D contains additional results for a more complex dataset with unequal element weights and sparser overlays, as well as a comparison to an additional local search baseline (hill climbing with restarts).

## 6.2 COVID-19 VACCINE DESIGN USING MAXIMUM $n$-TIMES COVERAGE

We used both the NTIMES-ILP and MARGINALGREEDY algorithms to design peptide vaccines for COVID-19 and evaluated the population coverage with different number of HLA-peptide hits ($1 \leq n \leq 8$) using EvalVax-Robust. For MARGINALGREEDY the time complexity of the algorithm is polynomial, $O(vbdp)$, where $v$ is the vaccine size, $b$ is the beam width, $d$ is the number of HLA diploid genotypes, and $p$ is the number of peptides. For the COVID-19 vaccine problem, this is $O(10^{12})$, and our implementation typically takes 225 CPU hours (with parallelization it finishes in <1 hour) for vaccine design. By point of contrast, exhaustive search is $\binom{p}{v}d$, which is $O(10^{55})$.

We compared our vaccine designs with 29 vaccine designs in the literature on the probability that a vaccine has at least $N$ peptide-HLA hits per individual in a population, and the expected number of per individual peptide-HLA hits in the population, which provides insight on how well a vaccine is displayed on average. Figure 3 shows the comparison between OptiVax (A) MHC class I and (B) class II vaccine designs at all vaccine sizes from 1–45 peptides (colored curves) and baseline vaccines

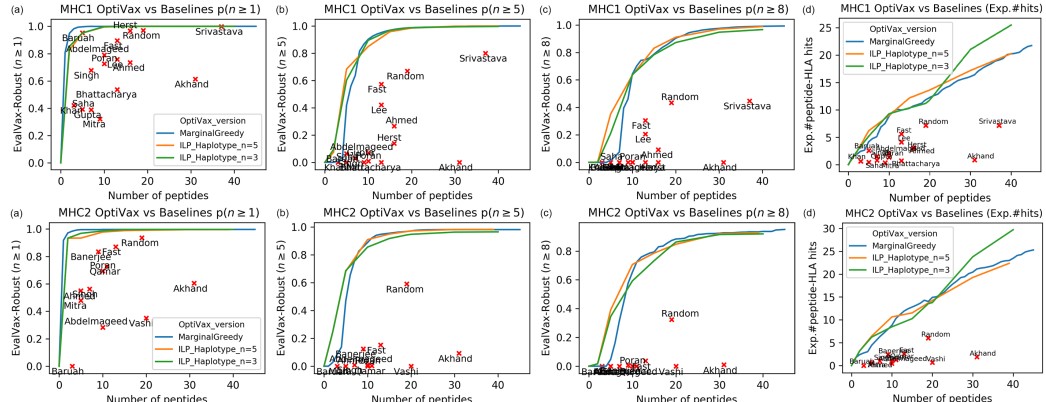

Figure 3: EvalVax population coverage evaluation of SARS-CoV-2 vaccines for (top) MHC class I and (bottom) MHC class II. (a) EvalVax-Robust population coverage with $n \geq 1$ peptide-HLA hits per individual, performances of 3 variants of OptiVax are shown by curves and baseline performance is shown by red crosses (labeled by name of first author). (b) EvalVax-Robust population coverage with $n \geq 5$ peptide-HLA hits. (c) EvalVax-Robust population coverage with $n \geq 8$ peptide-HLA hits. (d) Comparison of OptiVax and baselines on expected number of peptide-HLA hits.

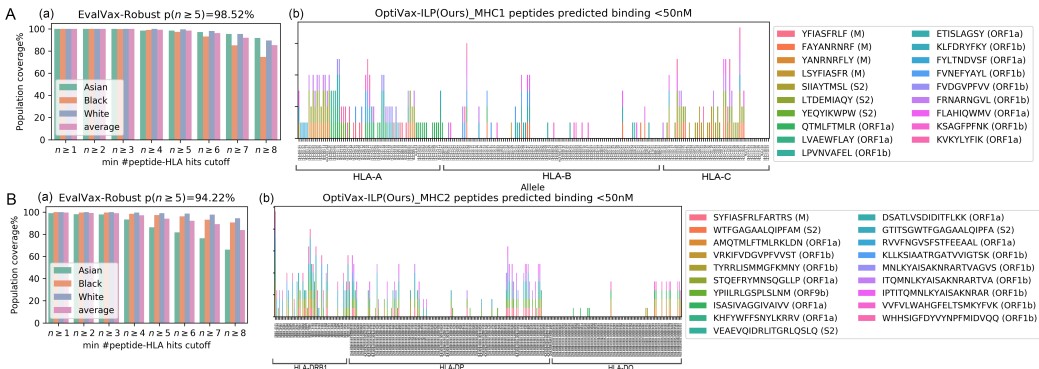

Figure 4: SARS-CoV-2 OptiVax selected peptide vaccine using NTIMES-ILP on haplotypes for (A) MHC class I and (B) MHC class II. (a) EvalVax-Robust population coverage at different per-individual number of peptide-HLA hit cutoffs for populations self-reporting as having White, Black, or Asian ancestry and average values. (b) Binding of vaccine peptides to each of the available alleles.

(red crosses) from prior work. We observe superior performance of OptiVax vaccine designs on all evaluation metrics at all vaccine sizes for both MHC class I and class II. Most baselines achieve reasonable coverage at $n \geq 1$ peptide hits. However, many fail to show a high probability of higher hit counts, indicating a lack of predicted redundancy if a single peptide is not displayed. Note that OptiVax-MarginalGreedy also outperforms all baselines on $n = 1$ coverage and achieve 99.99% (MHC class I) and 99.67% (MHC class II) coverage for $n = 1$, suggesting its capability to cover almost full population at least once while optimizing for higher peptide display diversity.

**MHC class I results.** We selected an optimized set of peptides from all SARS-CoV-2 proteins using NTIMES-ILP and MARGINALGREEDY. We use an MHC class I integrated model of peptide-HLA immunogenicity for our objective function. After all filtering steps, we considered 1,100 candidate peptides. With OptiVax-ILP, we design a vaccine with 19 peptides that achieves 99.999% EvalVax-Robust coverage over populations self-reporting as having Asian, Black, or White ancestry with at least one peptide-HLA hit per individual. This set of peptides also provides 98.52% coverage with at least 5 peptide-HLA hits and 85.40% coverage with at least 8 peptide-HLA hits (Figure 4A, Table S2). With OptiVax-MarginalGreedy, we design a vaccine with 19 peptides that achieves 99.999% EvalVax-Robust coverage over populations self-reporting as having Asian, Black, or White ancestry with at least one peptide-HLA hit per individual. This set of peptides also provides 98.58%

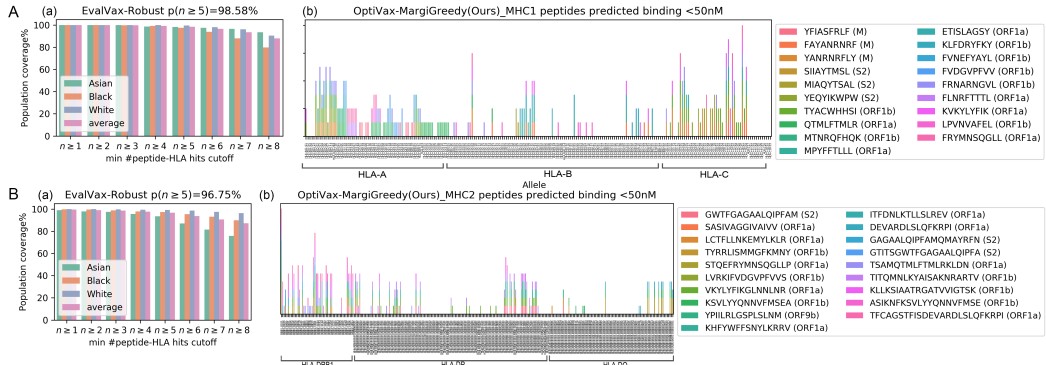

Figure 5: SARS-CoV-2 OptiVax selected peptide vaccine set using MARGINALGREEDY.

coverage with at least 5 peptide-HLA hits and 87.97% coverage with at least 8 peptide-HLA hits (Figure 5A, Table S2). The population level distribution of the number of peptide-HLA hits in White, Black, and Asian populations is shown in Figure 5A, where the expected number of peptide-HLA hits 11.355, 11.151 and 12.984, respectively.

**MHC class II results.** We use an MHC class II model of peptide-HLA immunogenicity for our objective function. After all filtering steps, we considered 4,195 candidate peptides. With NTIMES-ILP, we design a vaccine with 19 peptides that achieves 99.65% EvalVax-Robust coverage with at least one peptide-HLA hit per individual. This set of peptides also provides 94.22% coverage with at least 5 peptide-HLA hits and 83.76% coverage with at least 8 peptide-HLA hits (Figure 4B, Table S2). With MARGINALGREEDY, we design a vaccine with 19 peptides that achieves 99.63% EvalVax-Robust coverage with at least one peptide-HLA hit per individual. This set of peptides also provides 96.75% coverage with at least 5 peptide-HLA hits and 87.35% coverage with at least 8 peptide-HLA hits (Figure 4B, Table S2). The population level distribution of the number of peptide-HLA hits per individual in populations self-reporting as having Asian, Black, or White ancestry is shown in Figure 5B, where the expected peptide-HLA hits is 17.206, 16.085 and 11.210, respectively.

Table S2 shows the evaluation of our OptiVax-Robust vaccine designs using the MARGINALGREEDY algorithm compared to 29 designs by others as baselines. We note that it is natural that our designs that were both optimized and evaluated with the same objective performed the best. To provide a fair comparison, we also evaluated all designs with an immunogenicity model that does not incorporate clinical data and found that our designs also performed the best (Figure S1). The metric used in all cases is vaccine population coverage, which is a common metric (Bui et al., 2006). Thus, part of the contribution of the present work is emphasizing the value of combining machine learning predictions with combinatorial optimization for principled vaccine design.

## 7 CONCLUSION

We introduced the maximum $n$-times coverage problem, and showed that it is not submodular. We presented both a novel ILP based method and a beam search algorithm for solving the problem, and used them to produce a peptide vaccine design for COVID-19. We compared the resulting optimized peptide vaccine designs with 29 other published designs and found that the optimized designs provide substantially better population coverage for both MHC class I and class II presentation of viral peptides. The use of $n$-times coverage as an objective increases vaccine redundancy and thus the probability that one of the presented peptides will be immunogenic and produce a T cell response. We are presently testing our pan-strain COVID-19 vaccine in mouse challenge studies for protection against a COVID variant of concern. Our pan-strain vaccine delivers MHC class I and class II $n$-times coverage optimized epitopes with a single mRNA-LNP construct. Our methods can also be used to augment existing vaccine designs with peptides to improve their predicted population coverage by initializing the algorithm with peptides that are present in an existing design. We provide an open-source implementation of our methods at https://github.com/gifford-lab/optivax.

ACKNOWLEDGMENTS

This work was supported in part by Schmidt Futures, a Google Cloud Platform grant, and a C3.AI DTI Award to D.K.G. Ge Liu's contribution was made prior to joining Amazon.

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

## A    PROOF THAT $n$-TIMES COVERAGE IS NOT SUBMODULAR

**Theorem 2.** *The $n$-times coverage function $f_n(O)$ is not submodular.*

*Proof.* We show $f_n(O)$ is not submodular for $n = 2$ (a similar counter example can be found for any $n > 1$). Consider the example overlays $A$ in Table S1. When $n = 2$, none of $\{A_1\},\{A_2\},\{A_3\}$ or $\{A_2, A_3\}$ provides $n$-times coverage while $\{A_1, A_3\}$ covers $X_2$ two times and $\{A_1, A_2\}$ covers $X_4$ two times. Therefore, the marginal gain of adding $A_1$, $A_2$, or $A_3$ into an empty set is always zero, whereas adding $A_1$ into $\{A_2\}$ or $\{A_3\}$ achieves non-zero gains and adding $A_1$ into $\{A_2, A_3\}$ achieves even higher gain:

$$\Delta_f(A_1|\emptyset) := f_{n=2}(\{A_1\}) - f_{n=2}(\emptyset)$$
$$= 0$$
$$\Delta_f(A_1|\{A_2\}) := f_{n=2}(\{A_1, A_2\}) - f_{n=2}(\{A_2\})$$
$$= w(X_4) - 0 = 0.48$$
$$\Delta_f(A_1|\{A_3\}) := f_{n=2}(\{A_1, A_3\}) - f_{n=2}(\{A_3\})$$
$$= w(X_2) - 0 = 0.01$$
$$\Delta_f(A_1|\{A_2, A_3\}) := f_{n=2}(\{A_1, A_2, A_3\}) - f_{n=2}(\{A_2, A_3\})$$
$$= w(X_2) + w(X_4) - 0 = 0.49$$

Given that $\{A_3\} \subseteq \{A_2, A_3\}$ and $\Delta_f(A_1|\{A_3\}) < \Delta_f(A_1|\{A_2, A_3\})$, the function $f_{n=2}(O)$ does not satisfy the diminishing return property (Definition 2) and thus is not submodular.    □

Table S1: Coverage map of overlays used in the counter example.

| $w(X_i)$ | 0.01 | 0.01 | 0.5 | 0.48 |
|---|---|---|---|---|
|  | $X_1$ | $X_2$ | $X_3$ | $X_4$ |
| $A_1$ | 0 | 1 | 0 | 1 |
| $A_2$ | 0 | 0 | 1 | 1 |
| $A_3$ | 1 | 1 | 0 | 0 |

**Definition 2.** *(Submodularity) A function $f : 2^V \to \mathbb{R}$ is submodular if for every $A \subseteq B \subseteq V$ and $e \in V \setminus B$ it holds that $\Delta_f(e|A) \geq \Delta_f(e|B)$.*

Equivalently, $f$ is a submodular function if for every $A, B \subseteq v$, $f(A \cap B) + f(A \cup B) \leq f(A) + f(B)$.

## B    DETAILS OF VACCINE DESIGN

We score peptide-HLA immunogenicity based upon clinical data from convalescent COVID-19 patients (Section 5). For peptide-HLA combinations not observed in our clinical data we selected a machine learning model of immunogenicity that best predicted the peptide-HLA combinations we did observe. Our selected MHC class I model predicts a peptide-HLA combination will be immunogenic if they are predicted to bind with an affinity of $\leq 50$ nM by the mean of the predictions from PUFFIN (Zeng & Gifford, 2019), NetMHCpan-4.0 (Jurtz et al., 2017) and MHCflurry 2.0 (O'Donnell et al., 2020). Our selected MHC class II model predicts a peptide-HLA combination will be immunogenic if they are predicted to bind with an affinity of $\leq 50$ nM by NetMHCIIpan-4.0.

Peptides of length 8–10 residues can bind to HLA class I molecules whereas those of length 13–25 bind to HLA class II molecules (Rist et al., 2013; Chicz et al., 1992). We obtained the SARS-CoV-2 viral proteome from the first documented case from GISAID (Elbe & Buckland-Merrett, 2017) (sequence entry Wuhan/IPBCAMS-WH-01/2019). We applied sliding windows of length 8–10 (MHC class I) and 13–25 (MHC class II) to identify candidate peptides for inclusion in our peptide vaccine, resulting in 29,403 candidate peptides for MHC class I and 125,593 candidate peptides for MHC class II.

**Candidate peptide filtering.**   For our SARS-CoV-2 peptide vaccine design, we eliminate peptides that are expected to mutate and thus cause vaccine escape, peptides crossing cleavage sites, peptides that may be glycosylated, and peptides that are identical to peptides in the human proteome.

**Removal of mutable peptides.**   We eliminate peptides that are observed to mutate above an input threshold rate (0.002) to improve coverage over all SARS-CoV-2 variants and reduce the chance that the virus will mutate and escape vaccine-induced immunity in the future. When possible, we select peptides that are observed to be perfectly conserved across all observed SARS-CoV-2 viral genomes. Peptides that are observed to be perfectly conserved in thousands of examples may be functionally constrained to evolve slowly or not at all.

For SARS-CoV-2, we obtained the most up to date version of the GISAID database (Elbe & Buckland-Merrett, 2017) (as of 4:04pm EST February 4, 2021) and used Nextstrain (Hadfield et al., 2018)[1] to remove genomes with sequencing errors, translate the genome into proteins, and perform multiple sequence alignments (MSAs). We retrieved 451,198 sequences from GISAID, and 426,072 remained after Nextstrain quality processing. After quality processing, Nextstrain randomly sampled 34 genomes from every geographic region and month to produce a representative set of 12,884 genomes for evolutionary analysis. Nextstrain definition of a "region" can vary from a city (e.g., "Shanghai") to a larger geographical district. Spatial and temporal sampling in Nextstrain is designed to provide a representative sampling of sequences around the world.

The 12,884 genomes sampled by Nextstrain were then translated into protein sequences and aligned. We eliminated viral genome sequences that had a stop codon, a gap, an unknown amino acid (because of an uncalled nucleotide in the codon), or had a gene that lacked a starting methionine, except for ORF1b which does not begin with a methionine. This left a total of 12,789 sequences that were used to compute peptide level mutation probabilities. For each peptide, the probability of mutation was computed as the number of non-reference peptide sequences observed divided by the total number of peptide sequences observed.

**Removal of cleavage regions.**   SARS-CoV-2 contains a number of post-translation cleavage sites in ORF1a and ORF1b that result in a number of nonstructural protein products. Cleavage sites for ORF1a and ORF1b were obtained from UniProt (Consortium, 2019) under entry P0DTD1. In addition, a furin-like cleavage site has been identified in the Spike protein (Wang et al., 2020; Coutard et al., 2020). This cleavage occurs before peptides are loaded in the endoplasmic reticulum for class I or endosomes for class II. Any peptide that spans any of these cleavage sites is removed from consideration. This removes 3,887 peptides out of the 163,796 we consider across windows 8–10 (class I) and 13–25 (class II) ($\sim$2.4%).

**Removal of glycosylated peptides.**   Glycosylation is a post-translational modification that involves the covalent attachment of carbohydrates to specific motifs on the surface of the protein. We eliminate all peptides that are predicted to have N-linked glycosylation as it can inhibit MHC class I peptide loading and T cell recognition of peptides (Wolfert & Boons, 2013). We identified peptides that may be glycosylated with the NetNGlyc N-glycosylation prediction server (Gupta et al., 2004) and eliminated peptides where NetNGlyc predicted a non-zero N-glycosylation probability in any residue. This resulted in the elimination of 18,957 of the 154,996 peptides considered ($\sim$12%).

**Self-epitope removal.**   T cells are selected to ignore peptides derived from the normal human proteome, and thus we remove any self peptides from consideration for a vaccine. In addition, it is possible that a vaccine might stimulate the adaptive immune system to react to a self peptide that was presented at an abnormally high level, which could lead to an autoimmune disorder. All peptides from SARS-CoV-2 were scanned against the entire human proteome downloaded from UniProt (Consortium, 2019) under Proteome ID UP000005640. A total of 48 exact peptide matches (46 8-mers, two 9-mers) were discovered and eliminated from consideration.

**Datasets.**   OptiVax (Liu et al., 2020) software was obtained from GitHub (`https://github.com/gifford-lab/optivax`) and is available under an MIT license. Models and haplotype frequencies (Liu et al., 2020) were obtained from Mendeley Data (`https://doi.org/10.17632/`

---

[1]from GitHub commit 639c63f25e0bf30c900f8d3d937de4063d96f791

`cfxkfy9zp4.1,https://doi.org/10.17632/gs8c2jpvdn.1`) and are available under a Creative Commons Attribution 4.0 International license.

**Computational resources.** We utilized Google Cloud Platform machines with 224 CPU cores for MarginalGreedy optimization and parallelized computation across all CPU cores. For ILP designs, we used our own computing resources with 8 CPU cores. The prediction of peptide-HLA binding with machine learning models (NetMHCpan, NetMHCIIpan, MHCflurry, PUFFIN) was done using our own computing resources with ∼200 CPU cores and NVIDIA GeForce RTX 2080 Ti GPUs.

**ILP solver.** We use the Python-MIP solver (Santos & Toffolo, 2020), which is a well-maintained Mixed Integer Program (MIP) Solver with a Python API. Other solvers, such as the SCIP (Achterberg, 2009) and the LMHS (Saikko et al., 2016) solvers were considered, yet were deemed to not be the best options for this problem, as they are optimized to solve more general problems than MIPs without any guarantees of better performance compared to the Python-MIP solver.

| Peptide Set | Vaccine Size | EvalVax-Robust $p(n \geq 1)$ | EvalVax-Robust $p(n \geq 5)$ | EvalVax-Robust $p(n \geq 8)$ | Exp. # peptide-HLA hits / vaccine size | Exp. # peptide-HLA hits (White) | Exp. # peptide-HLA hits (Black) | Exp. # peptide-HLA hits (Asian) |
|---|---|---|---|---|---|---|---|---|
| **MHC Class I Peptide Vaccine Evaluation** | | | | | | | | |
| S-protein | 3795 | 100.00% | 99.43% | 99.06% | 40.183 | 38.212 | 38.250 | 44.086 |
| S1-subunit | 2055 | 100.00% | 99.05% | 97.37% | 20.788 | 20.616 | 20.397 | 21.351 |
| OptiVax-MarginalGreedy(Ours) | 19 | 100.00% | 98.58% | 87.97% | 11.830 | 11.355 | 11.151 | 12.984 |
| OptiVax-ILP(Ours) | 19 | 100.00% | 98.52% | 85.40% | 11.164 | 10.883 | 10.430 | 12.180 |
| Srivastava et al. (2020) | 37 | 99.90% | 80.06% | 44.74% | 7.163 | 7.533 | 7.018 | 6.938 |
| Random subset of binders | 19 | 97.09% | 67.00% | 43.42% | 7.131 | 7.368 | 6.229 | 7.796 |
| Fast et al. (2020) | 13 | 89.60% | 57.38% | 30.55% | 5.558 | 5.616 | 4.386 | 6.672 |
| Herst et al. (2020) | 52 | 97.76% | 50.93% | 13.50% | 4.719 | 5.207 | 4.473 | 4.477 |
| Lee & Koohy (2020) | 13 | 75.78% | 42.18% | 20.60% | 4.113 | 4.397 | 3.384 | 4.558 |
| Ahmed et al. (2020) | 16 | 73.49% | 26.65% | 9.29% | 3.085 | 3.051 | 2.248 | 3.957 |
| Herst et al. (2020) | 16 | 96.70% | 13.97% | 0.07% | 2.931 | 3.236 | 2.988 | 2.568 |
| Abdelmageed et al. (2020) | 10 | 79.09% | 7.05% | 0.11% | 2.000 | 2.192 | 1.777 | 2.030 |
| Baruah & Bose (2020) | 5 | 95.31% | 6.39% | 0.00% | 2.565 | 3.128 | 2.031 | 2.537 |
| Gupta et al. (2020) | 7 | 38.91% | 3.01% | 0.00% | 0.804 | 0.747 | 0.388 | 1.278 |
| Singh et al. (2020) | 7 | 67.80% | 2.84% | 0.00% | 1.499 | 1.431 | 1.305 | 1.760 |
| Vashi et al. (2020) | 51 | 84.86% | 2.53% | 0.00% | 1.802 | 1.976 | 1.711 | 1.720 |
| Poran et al. (2020) | 10 | 72.56% | 0.84% | 0.00% | 1.355 | 0.997 | 1.383 | 1.686 |
| Khan et al. (2020) | 3 | 42.08% | 0.09% | 0.00% | 0.614 | 0.702 | 0.878 | 0.263 |
| Akhand et al. (2020) | 31 | 61.44% | 0.04% | 0.00% | 0.840 | 1.062 | 0.774 | 0.682 |
| Bhattacharya et al. (2020) | 13 | 53.78% | 0.01% | 0.00% | 0.710 | 0.977 | 0.635 | 0.518 |
| Saha & Prasad (2020) | 5 | 39.30% | 0.00% | 0.00% | 0.423 | 0.496 | 0.291 | 0.482 |
| Mitra et al. (2020) | 9 | 32.13% | 0.00% | 0.00% | 0.355 | 0.504 | 0.251 | 0.311 |
| **MHC Class II Peptide Vaccine Evaluation** | | | | | | | | |
| OptiVax-MarginalGreedy(Ours) | 19 | 99.63% | 96.75% | 87.35% | 14.017 | 16.989 | 14.375 | 10.686 |
| S-protein | 16315 | 99.28% | 96.53% | 96.04% | 395.096 | 541.429 | 416.406 | 227.455 |
| OptiVax-ILP(Ours) | 19 | 99.65% | 94.22% | 83.76% | 13.540 | 15.900 | 14.762 | 9.958 |
| S1-subunit | 8905 | 98.86% | 91.87% | 90.97% | 177.838 | 265.183 | 175.615 | 92.717 |
| Ramaiah & Arumugaswami (2020) | 134 | 98.88% | 90.20% | 83.97% | 33.743 | 45.044 | 38.254 | 17.932 |
| Random subset of binders | 19 | 93.70% | 59.13% | 32.47% | 6.033 | 7.822 | 6.527 | 3.750 |
| Fast et al. (2020) | 13 | 86.99% | 15.24% | 3.69% | 2.560 | 3.650 | 2.262 | 1.769 |
| Banerjee et al. (2020) | 9 | 83.51% | 12.49% | 0.66% | 2.398 | 3.162 | 2.354 | 1.679 |
| Akhand et al. (2020) | 31 | 60.45% | 9.22% | 1.01% | 1.886 | 2.531 | 2.536 | 0.591 |
| Singh et al. (2020) | 7 | 56.29% | 0.96% | 0.00% | 0.981 | 1.438 | 1.113 | 0.392 |
| Abdelmageed et al. (2020) | 10 | 28.40% | 0.96% | 0.00% | 0.479 | 0.919 | 0.274 | 0.244 |
| Tahir ul Qamar et al. (2020) | 11 | 72.75% | 0.27% | 0.00% | 1.278 | 1.840 | 1.464 | 0.530 |
| Mitra et al. (2020) | 5 | 47.92% | 0.04% | 0.00% | 0.657 | 0.905 | 0.579 | 0.488 |
| Vashi et al. (2020) | 20 | 35.12% | 0.04% | 0.00% | 0.673 | 0.959 | 0.618 | 0.442 |
| Poran et al. (2020) | 10 | 69.37% | 0.00% | 0.00% | 0.983 | 1.469 | 0.910 | 0.569 |
| Ahmed et al. (2020) | 5 | 54.96% | 0.00% | 0.00% | 0.654 | 0.736 | 0.717 | 0.510 |
| Baruah & Bose (2020) | 3 | 0.00% | 0.00% | 0.00% | 0.000 | 0.000 | 0.000 | 0.000 |

Table S2: Comparison of existing baselines, S-protein peptides, and OptiVax-Robust peptide vaccine designs on various population coverage evaluation metrics. The rows are sorted by EvalVax-Robust $p(n \geq 1)$. Random subsets are generated 200 times. The binders used for generating random subsets are defined as peptides that are predicted to bind with $\leq$ 50 nM to more than 5 of the alleles.

## C  EVALUATION OF OPTIVAX-ROBUST VACCINE DESIGN AND BASELINES WITH NON-EXPERIMENTALLY-CALIBRATED MACHINE LEARNING PREDICTIONS

We found that OPTIVAX produced vaccine designs superior to the baselines we tested even when it used immunogenicity models that did not rely upon clinical data of peptide immunogenicity. Since our immunogenicity model used experimental data that was not accessible to the baseline methods, our designs have an advantage over baseline vaccines that did not use calibrated machine learning predictions. We repeated vaccine design using an ensemble of NetMHCpan-4.0 and MHCflurry with 50 nM predicted affinity cutoff for predicting MHC class I immunogenicity and NetMHCIIpan-4.0 for MHC class II. As shown in Figure S1, OptiVax-Robust again shows superior performance on all metrics at all vaccine size under 35, indicating the success of MARGINALGREEDY in optimizing population coverage with diverse peptide display.

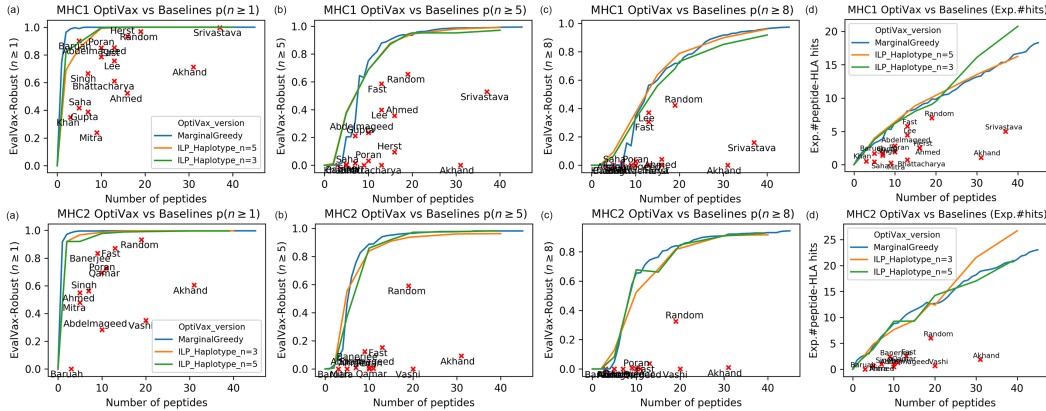

Figure S1: EvalVax population coverage evaluation of SARS-CoV-2 vaccines for (top) MHC class I and (bottom) MHC class II using non-experimentally calibrated machine learning predictions. (a) EvalVax-Robust population coverage with $n \geq 1$ peptide-HLA hits per individual, performances of 3 variants of OptiVax are shown by curves and baseline performance is shown by red crosses (labeled by name of first author). (b) EvalVax-Robust population coverage with $n \geq 5$ peptide-HLA hits. (c) EvalVax-Robust population coverage with $n \geq 8$ peptide-HLA hits. (d) Comparison of OptiVax and baselines on expected number of peptide-HLA hits.

## D  ADDITIONAL EXPERIMENTS ON A DATASET WITH UNEQUAL WEIGHTS AND SPARSE COVERAGE

We further expanded our evaluation to include a hill climbing based method and a more complex dataset. We generated a dataset named LARGE with sparse overlay coverage and unequal element weights to provide a further basis of comparison for $n$-times coverage methods. The LARGE dataset was created by generating a set of 30 overlays to cover a set of 10 elements with unequal weights, and the probability of covering an element 0, 1, or 2 times is 0.6, 0.3, and 0.1, respectively. We added an additional baseline method using hill climbing with random restarts for comparison with other methods. For hill climbing we used stochastic local search that queries a randomly selected subset of neighbors (Algorithm 2). We used stochastic local search because for a solution size of $k$, a single exhaustive local search step requires querying all possible solutions that are 1 peptide different from the current solution, which may result in $k \times$ #_total candidate_peptide queries to the oracle that computes $n$-times population coverage. This approach is too expensive in typical vaccine design problems with thousands of candidate peptides. For a fair comparison we have set the number of restarts and hill climbing steps such that the total number of oracle queries needed to build a hill climbing solution of size $k$ is equivalent to that of MarginalGreedy. Figure S2 shows that hill climbing is effective when the solution size is small, and but fails to reach 100% coverage when MarginalGreedy reaches 100% coverage. MarginalGreedy and True Optimal reach 100% coverage

with approximately the same number of overlays. Our integer programming $n$-times coverage formulation achieves exact true optimal results on the LARGE dataset.

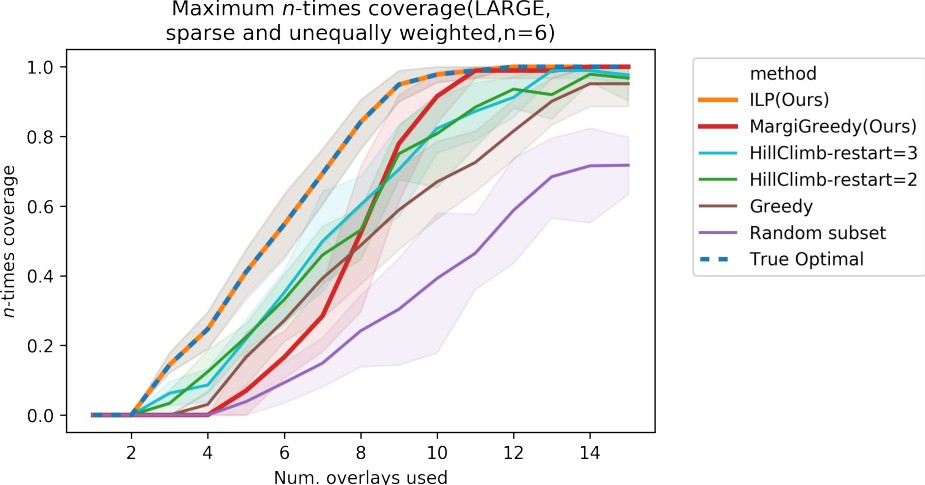

Figure S2: Comparison of $n$-times coverage methods on the LARGE dataset that includes hill climbing (Algorithm 2). The LARGE dataset contains unequal element weights and sparse coverage.

---

**Algorithm 2** HILLCLIMB algorithm (in MAXIMUM $n$-TIMES COVERAGE with cardinality constraint)

---

**Input:** Weights of the elements in $\mathcal{X}$:$w(X_1), w(X_2), \ldots, w(X_l)$, ground set of overlays $\mathcal{A}$ where each overlay $j$ in $\mathcal{A}$ covers $X_i$ for $c_j(X_i)$ times, cardinality constraint $k$, target minimum # times being covered $n_{target}$, number of restarts $M$, number of hill climbing steps $T$, number of neighbors considered in one hill climb step $N$.
Initialize solution set $\mathcal{S} = \emptyset$
Using $n_{target}$-times coverage function $f(O) = \sum_{i=1}^{l} w(X_i)\mathbb{1}_{\{\sum_{j \in O} c_j(X_i) \geq n_{target}\}}$ as objective.
**for** $m = 1, \ldots, M$ **do**
    Randomly select a set of $k$ overlays from $\mathcal{A}$ as starting solution $O^*$, compute $\tau_{best} = f(O^*)$ as the best coverage so far.
    **for** $step = 1, \ldots, T$ **do**
        Randomly pick an overlay $a$ from $O^*$ as candidate to be replaced
        Randomly select $N$ overlays from currently unselected set $\mathcal{N} \subseteq \mathcal{A} \setminus O$
        **for** $a' \in \mathcal{N}$ **do**                      ▷ Loop over the selected neighbors
            **if** $f(O^* \setminus \{a\} \cup \{a'\}) > \tau_{best}$ **then**
                $O^* \leftarrow O^* \setminus \{a\} \cup \{a'\}$          ▷ Replace the overlay with a better one
                $\tau_{best} \leftarrow f(O^*)$
    $\mathcal{S} \leftarrow \mathcal{S} \cup \{O^*\}$          ▷ Add the best solution in current run into solution pool
    $O* = \arg\max_{O \in \mathcal{S}} f(O)$             ▷ Use the best result as final solution
**Output:** The final selected subset of overlays $O^*$

---

## E   HIGH-RESOLUTION FIGURES FOR VACCINE MHC HITS

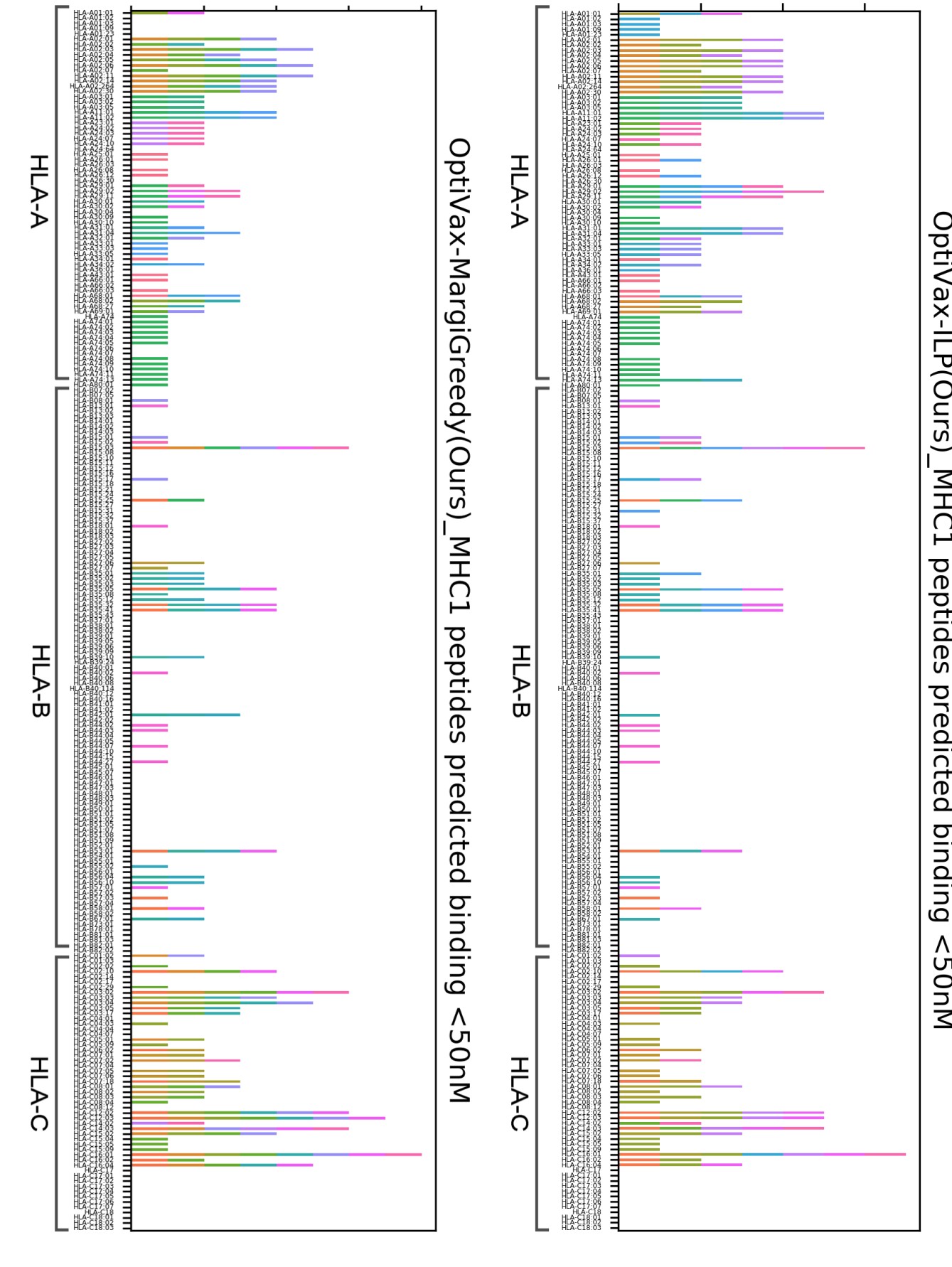

Figure S3: High-resolution MHC class I binding plots from Figure 4 and Figure 5.

