# OpenReview forum: "Maximum n-times Coverage for Vaccine Design"
_ICLR.cc/2022/Conference — ICLR 2022 Poster_

### Official Review · Reviewer_csaP · 2021-11-02

**Correctness:** 3
**Technical Novelty And Significance:** 3
**Empirical Novelty And Significance:** 3
**Recommendation:** 6
**Confidence:** 3

**Main Review:**

As far as algorithms are concerned, the greedy approach is standard for set-covering type problems, even though it seems that it has not been used for this particular set covering version before. Most of the ILP formulation seems standard as well, however, it is not clear how the objective function may be linearized and this requires additional work and Big-M-constraints.

The vaccine design process seems sound, building on published work. Experimental results show that the ILP comes close close to the optimum solution. Finally, the resulting vaccine designs are succcessfully compared to designs from the literature.

As a minor comment, I did not fully understand the pseudo-code for the beam search: Initally, B^0 is empty, so the beam search (outer for-loop) iterates over an empty set and will not build a candidate set. Wouldn't then B^{t+1} be empty as well?

**Summary Of The Paper:**

The submission is concerned with vaccine design and proposes to model the task as a combinatorial optimization problem that is essentially a generalization of set cover. The authors describe two algorithms: A marginally greedy algorithm using beam search and a mixed integer linear programming approach.

More precisely, the combinatorial problem consists in covering weighted elements (the population) with overlays (peptides) while maximizing the total weight of those elements that are covered at least n times. Each overlay may only be used once, but may cover the same element multiple times. In total, no more than k overlays may be chosen.

Additionally, the submission features a machine learning based tool that predicts the effectiveness of the designed vaccine.

**Summary Of The Review:**

The submission is built with a clear application in mind. It provides two reasonable approaches for a well-modeled problem. There are little theoretical contributions, but I found the practical contribution well-executed and convincing.

---

> ### Author Response · Authors · 2021-11-20
> **Response to Reviewer csaP**
>
> We thank the reviewer for the feedback and interest in our work.
>
> **[ILP objective function]**  We thank the reviewer for this comment and have clarified the paper accordingly.  The objective function presented in Section 3 of the paper is linear and it is fed directly into the ILP solver to produce the results shown in the paper.   We introduce the Big-M method to linearize the objective function, which is a step-function. The helper binary variables $t$ are introduced in the linearization process. After the ILP has run, if $t_j=1$, then the element $X_j$ has been covered at least $n$ times, and conversely if $t_j=0$, then the element $X_j$ has not been covered at least $n$ times. We further formulate the inequalities necessary to ensure that the above relationship holds. Finally, we write the objective function using the helper variables  $t$: $f_n(O) = \sum_{j=1, \dots, l} t_j w(X_j)$. This objective function is the sum over all the elements $X_j$ of the helper variable $t_j$ of every element multiplied by the frequency/weight of that element. As the function is a sum over $l$ terms of a binary variable times a constant, it is linear. We have clarified this point in Section 3 in the paper.
>
> **[Pseudocode]**  We agree that this is confusing and we have added a comment to Algorithm 1 to avoid this misunderstanding.  Please note that the inner loop of Algorithm 1 builds $K^t$ that is $\\{\\{O \cup a_1\\}, \\{O \cup a_2\\}, \dots \\{O \cup a_n\\}\\}$ where $a_1, \dots, a_n$ are all possible overlays that are not in the current candidate solution $O$.   We then select the top b of these candidate solutions to become the new $B^{t+1}$.    Thus at the outset when $O$ is empty, $B^{1}$ will consist of the top $b$ candidate solutions as scored by the objective function $f(S)$ where the solutions are each single overlays $a_1, \dots, a_n$.  Each iteration will update $B$ with the top $b$ candidate solutions.

---

### Official Review · Reviewer_4qAy · 2021-11-02

**Correctness:** 3
**Technical Novelty And Significance:** 3
**Empirical Novelty And Significance:** 2
**Recommendation:** 6
**Confidence:** 3

**Main Review:**

The new problem appears to be interesting. I am not aware of a problem with such a structure, although I would not be surprised if there were previous studies on this framework in the area of combinatorial design or also in one of the several variants of group testing.
The experimental evaluation of the approach in on a currently hype area is also valuable.
There are a couple of concerns I have:
- the pseudocode for algorithm 1 is not clear to me. It seems that the two inner for loop will always end up constructing K^t = A.
- in the empirical study on the toy examples, the weights are not used, which is strange since it is what makes the problem different from the  classical variants of set cover. It is also not clear to me from the description where the weights enter in the evaluations of the design for covid 19. The authors only mention coverage and it is not clear whether that is meant in terms of sum of weights or what else.
Due to these concerns I am not giving a fully acceptance to the paper but I am willing to receonsider my score on the basis of the authors' reply.

Minor problems: the figures are very small and it is very difficult to appreaciate on paper what they show.

**Summary Of The Paper:**

The paper introduces a variant/generalization of multi set multi cover problem, where the aim is to maximize the weight of the elements covered at least n times by up to k overlays (subsets of a given input familiy of sets over the elements' universe). The authors show that the objective function is not submodular, hence does not admit a classical greedy approach. They show an ILP formulation that provides an optimal solution but not efficiently. They propose a greedy approach based on maiximization of marginal gains endowed with a look-ahead tie breaking that should prevent failure to attain n-coverage because of wrong initial choices guided by the sole marginal gain maximization objective.
The problem is meant to model vaccine design and show empirical evaluation on two toy data sets and presenting comparisons between the greedy approach and the ILP approach for a covid 19 vaccine.

**Summary Of The Review:**

Strenght: The paper offers an interesting and apparently unexplored variant of set cover that can model a vaccin design problem. The evaluation seem to show good performance.
Weaknesses: there seems to be something unclear in the pseudocode of the greedy algorithm that prevents me from fully assess its behaviour; the importance of the weights in the experiments is not clearly discussed (without this, the problem reduces to known Multi-set multi cover)

---

> ### Author Response · Authors · 2021-11-20
> **Response to Reviewer 4qAy**
>
> We thank the reviewer for the feedback and interest in our work.
>
> **[Novelty]**  We thank the reviewer for their comments, and we clarify how our vaccine design approach substantially differs from prior work in our revised paper. We frame vaccine design as maximum n-times coverage with the goal that each vaccinated individual in a population will be “covered” by multiple immunogenic peptides. While it might be assumed that an individual will be immunized if they display a single peptide, three independent lines of reasoning support the need for n-times coverage:
>
> 1. When an individual displays multiple peptides their immune system activates and expands more than one set of T cell clonotypes that are poised to fight viral infection (Sekine et al., 2020; Schultheiß et al., 2020; Grifoni et al., 2020).
> 2. The peptides that are immunogenic vary from one individual to another, and thus having multiple peptides displayed increases the probability at least one will be strongly immunogenic (Croft et al., 2019).
> 3. If a virus evolves and changes its peptide composition, using multiple peptides reduces the chance of viral escape (Wibmer et al., 2021).
>
> Thus an n-times coverage constraint is strongly motivated by vaccine immunology. An n-times coverage constraint has not been considered by previous work and the constraint results in a novel optimization task with no previous solution.
>
> **[Pseudocode]**  We agree this is confusing and we have added a comment to Algorithm 1 to avoid this misunderstanding.  Please note that the inner expression of the loop does not union $K^t$, $O$, and $\{a\}$ together.  If it did, it would be the case that $K^t = A$ as you observe.   The subtlety here is the inclusion of the “$\\{$” and “$\\}$” brackets to indicate set construction.   As written $K^t$ is a set of sets, and the inner loop expression adds one more set that consists of the candidate solution $O$ combined with $a$.    Thus the inner loop construction produces the set $K^t$ that consists of $\\{\\{O \cup a_1\\}, \\{O \cup a_2\\}, \dots, \\{O \cup a_n\\}\\}$ where $a_1, \dots, a_n$ are all possible overlays that are not in the current candidate solution $O$.
>
> **[Empirical study and weights]**  We conducted new experiments with unequal weights and unequal overlay coverage probabilities to make the multicover problem more challenging (Appendix D).  Figure S2 shows that our MarginalGreedy solution achieves top performance for high-population coverage and our ILP solution achieves true optimal performance.
>
> We have revised the paper to improve the last paragraph of Section 5 that describes the connection between vaccine design and $n$-times coverage.  An individual’s genotype is analogous to an element in the multi-set multi-cover problem, and each candidate peptide is analogous to an overlay which provides a varying number of hits (number of times covered) to each genotype.  The weights are the frequencies of different genotypes (how often they appear in a population), and thus maximizing the sum of frequencies of genotypes that are covered by at least $n$ peptides maximizes $n$-times population coverage.
>
> **[Figures]**  We have provided higher-resolution figures in Appendix E.

---

### Official Review · Reviewer_Hy3s · 2021-11-04

**Correctness:** 2
**Technical Novelty And Significance:** 3
**Empirical Novelty And Significance:** 2
**Recommendation:** 6
**Confidence:** 4

**Main Review:**

The novelty of the proposed work is limited. A large number of existing approaches have formulated element-based vaccine design as a discrete optimization problem (e.g., [Malone et al., Scientific Reports 2020], [Theiler and Korder, Statistics in Medicine 2018], [Schulber and Kohlbacher, Genome Medicine 2016], [Oyarzun and Kobe, International Journal of Immunogenetics 2015], [Lundegaard et al., Proceedings of the $1^{st}$ ACM International Conference on Bioinformatics and Computational Biology 2010], [Toussaint et al., PLOS Computational Biology 2008]). While the proposed approach may have advantages compared to those formulations, the authors do not provide any context of how their formulation differs from the well-established approaches. So it is difficult to gauge the novelty of the proposed approach with respect to current approaches.

In terms of technical soundness, the theoretical contributions seem sound. I did not thoroughly verify the proofs of NP-completeness, but the argument seems sound. The greedy algorithm for addressing the optimization problem does not have any associated guarantees, but it also seems reasonable.

In terms of biology, the formulation is not very meaningful. For example, MHC (HLA in human) I leads to an entirely different immune response than MHC II (cytotoxic/cell killing for MHC I compared to recruiting various other immune cells for MHC II). A good T-cell vaccine should do both. Additionally, some haplotypes are more common than others, and the proposed approach ignores such pair-wise correlations. Thus, while it is clear that the formulated problem lends itself well to a set covering formulation, it is not clear to me that this formulation is meaningful.

Considering the experimental results, some obvious baselines are missing. For example, a greedy hill climbing with random restarts approach is a typical way to approach such discrete optimization problems, so it would be useful to show how much the proposed margin-based algorithm outperforms that. Further, since the exact solution to the problem can be formulated as an ILP, modern anytime solvers (which also offer guaranteed bounds), such as SCIP [Achterberg, Mathematical Programming Computation 2009] or LMHS [Saikko et al., SAT 2016], should be included in the comparison.

The paper is generally clear and well written. The references are inconsistently formatted. It would be helpful to increase the font size on the figures.

The reproducibility and provided resources of this work are in line with the field. An expert would likely be able to take the formulations in the paper and turn them into code.

**Summary Of The Paper:**

Update: I have read the author feedback and other reviews. Based on the more detailed comparison to existing methods, I have updated several of my scores.

---

In this work, the authors propose a variant of the set covering problem which aim to formalize peptide-based vaccine design as an optimization problem. The authors first formulate the problem and then draw connections to other set covering problems; they then show that the problem is NP-complete. They then propose a greedy algorithm to perform the optimization. A set of experiments suggests the proposed approach outperforms the formulations considered by the authors.

**Summary Of The Review:**

Considering the lack of theoretical novelty and biological accuracy, it is difficult for me to see the impact of the work as it is currently presented.

---

> ### Author Response · Authors · 2021-11-20
> **Response to Reviewer Hy3s - Part 1 of 2**
>
> We thank the reviewer for their comments, and we clarify how our vaccine design approach substantially differs from prior work in our revised paper.   We frame vaccine design as maximum n-times coverage with the goal that each vaccinated individual in a population will be “covered” by multiple immunogenic peptides. While it might be assumed that an individual will be immunized if they display a single peptide, three independent lines of reasoning support the need for n-times coverage:
>
> 1. When an individual displays multiple peptides their immune system activates and expands more than one set of T cell clonotypes that are poised to fight viral infection (Sekine et al., 2020; Schultheiß et al., 2020; Grifoni et al., 2020).
> 2. The peptides that are immunogenic vary from one individual to another, and thus having multiple peptides displayed increases the probability at least one will be strongly immunogenic (Croft et al., 2019).
> 3. If a virus evolves and changes its peptide composition, using multiple peptides reduces the chance of viral escape (Wibmer et al., 2021).
>
> Thus an n-times coverage constraint is strongly motivated by vaccine immunology.  An n-times coverage constraint has not been considered by previous work and the constraint results in a novel optimization task with no previous solution.
>
> **[Novelty]**  We thank the reviewer for these references, and in our revised paper we summarize the differences between maximum n-times coverage and these other formulations.   We agree that other work has formulated element-based vaccine design as a discrete optimization problem, but this previous work has considered simpler tasks that do not anticipate or generalize to n-times coverage.  Differences between our results and this previous work are as follows:
> * [Malone et al. 2020] states “We consider that a vaccine causes a positive response if at least one of its elements causes a positive response” and optimizes vaccine design for population coverage with a positive response.  Thus, Malone et al. provide 1-times coverage.  As described in our paper, n-times coverage provides at least n elements of positive response coverage for each person, which provides more T cell clonotypes to fight disease, accounts for variability of epitope immunogenicity, and counters viral drift.    This distinguishes the n-times coverage from Malone et al. and creates a novel optimization task to provide improved vaccine designs.
> * [Theiler and Korder 2018] consider the task of designing a single peptide sequence that covers a set of diverse but related input epitopes to accommodate antigenic drift.   This work does not consider population coverage or the associated optimization tasks.
> * [Schulber and Kohlbacher 2016] consider the task of designing spacers for string-of-beads peptide delivery.   This work does not consider optimization of vaccine population coverage or related tasks.
> * [Oyarzun and Kobe 2015] is a review article on vaccine design, and summarizes work as of 2015 on algorithms for optimizing vaccine population coverage.   The reviewed methods implement 1-times coverage, and thus do not consider redundancy for providing a diverse set of T cell clonotypes to fight disease, do not account for the variability of epitope immunogenicity, and do not account for viral drift.
> * [Lundegaard et al. 2010] consider the task of simultaneously optimizing for population and variant coverage with the iterative PopCover algorithm.  PopCover does not consider haplotypes, and is a heuristic algorithm that does not provide specific population coverage at any specific coverage constraint (1-times or n-times).
> * [Toussaint et al. 2008] formulates 1-times coverage of an ILP problem and does not consider haplotypes.
>
> This prior work has not considered formalizing the coverage of individuals with multiple epitopes with an n-times constraint and thus has produced solutions to the 1-times coverage task.  Existing solutions to 1-times coverage do not anticipate or solve the n-times coverage task.  Both Malone et al. (2020) and Toussaint et al.(2008) provide solutions to 1-times coverage, Lundegaard et al. (2010) does not provide specific population coverage guarantees, and Oyarzun & Kobe (2015) reviews methods for 1-times coverage.  Discrete optimization has been used for other aspects of vaccine design that are unrelated to population coverage, such designing a single peptide sequence that covers a set of diverse but related set of input epitopes (Theiler & Korber (2018)), and designing spacers for string-of-beads peptide delivery (Schubert & Kohlbacher (2016)).

---

> ### Author Response · Authors · 2021-11-20
> **Response to Reviewer Hy3s - Part 2 of 2**
>
> **[Biological formulation]**  In our revised paper we emphasize that we design vaccines that have both a MHC class I and class II component using n-times coverage, and thus our vaccines contain both of these important components to produce an effective immune response.
>
> As we have clarified in Section 5, our methods for n-times coverage vaccine design utilize the haplotype frequencies of 2,138 MHC class I haplotypes and 1,711 MHC class II haplotypes that describe MHC alleles that are inherited together.    A haplotype is defined as “a set of DNA variations, or polymorphisms, that tend to be inherited together” (https://www.genome.gov/genetics-glossary/haplotype) and thus comes from one parent.    Thus each individual has two haplotypes, one from each parent, that comprise their MHC diplotype.   We explicitly model pair-wise MHC allelic correlations (linkage disequilibrium) within haplotypes by our use of haplotype frequencies.   We make the conventional equilibrium assumption that diplotype frequencies are the product of haplotype frequencies (e.g., we model over 4 million class I diplotypes) and that the inheritance of haplotypes is independent from the pool of parents.
>
> **[Baselines]**  We thank the reviewer for mentioning additional baselines using local search (such as Hill Climbing) that start with random solutions and continue to improve it by searching on neighborhood solutions.   We now include these results in the paper as a point of comparison.  While this type of algorithm may work well on certain optimization problems, it has not yet proven to be effective on set cover problems (not to mention the even more difficult multiset multicover problem) where the neighborhood space is huge. As far as we know, there is no existing literature using Hill Climbing as a direct solution to the set cover problem, except for [Akhter, F. (2015)] which only uses Hill Climbing to further modify an existing solution created by a linear programming or greedy method.
>
> We addressed the reviewer's question with a new benchmark on a new LARGE dataset, and we include the details of the experiment and the Hill Climbing algorithm in Appendix D of the revised paper.  Note that for a solution size of $k$, a single exhaustive local search step requires querying all possible solutions that are 1 peptide different to current solution, which may result in k * #total_candidate_peptide queries to the oracle (n-times population coverage computation) and is too expensive in typical vaccine design problems with thousands of candidate peptides. Therefore we used stochastic local search that queries a randomly selected subset of neighbors. To make a fair comparison, we have set the number of restarts and hill climbing steps accordingly such that the total number of oracle queries needed to build a solution of size $k$ is equivalent to that of MarginalGreedy. As can be seen from Figure S2, Hill Climbing (similar to Greedy) is only effective when the solution size is small, and it fails to reach 100% coverage in most of the scenarios while MarginalGreedy reaches 100% almost at the same k with True optimal. Our integer programming $n$-times coverage formulation achieves exact true optimal results on this experiment.
>
> **[ILP solvers]** We thank the reviewer for these references and suggestions.  As the ILP problem is NP-Complete, we are unaware of any guaranteed bounds for any solver.  The solver we use, Python-MIP,  offers an estimate of how far the solution is from the optimal one as it runs, yet this is only an estimate.  Python-MIP is a widely used Mixed Integer Programming Solver with a well maintained codebase and a large user population (> 700,000 downloads).   Details on the performance and the inner-workings of the solver can be found in Santos, Haroldo G., and T. A. Toffolo. "Mixed Integer Linear Programming With Python." (2020).
>
> Our revised paper describes LHMS and SCIP as alternative ILP solvers.  The LMHS solver is primarily a maxSAT solver, which has the capability to also solve ILP as stated in the LMHS paper.   After carefully reviewing the work on LMHS we do not see evidence it will provide a superior solution when compared with Python-MIP, and carefully benchmarking ILP solvers is not the focus of the present work.   The SCIP solver is optimized to solve constraint integer programs.   As is stated in the abstract for the SCIP paper  “(we) show by computational experiments that SCIP is almost competitive to specialized commercial MIP solvers, even though SCIP supports the more general constraint integer programming paradigm”.  This statement suggests SCIP will be at best equivalent to Python-MIP for the present work’s ILP task.
>
> **[Formatting]** We have revised the citation format and provided higher-resolution figures in Appendix E.

---

> > ### Comment · Reviewer_Hy3s · 2021-11-22
> > **Updated scores**
> >
> > Thanks to the authors for the detailed response to my concerns. I have updated my scores to reflect the novelty with respect to existing work.

---

> > > ### Author Response · Authors · 2021-11-22
> > > **Thank you Reviewer Hy3s / Ancillary scores**
> > >
> > > Thank you for your reply and updated score.  If you believe we have addressed your previous concerns about correctness and novelty, we would greatly appreciate your also revising the respective scores.

---

> > > > ### Comment · Reviewer_Hy3s · 2021-11-23
> > > > **Re: Ancillary scores**
> > > >
> > > > Hi,
> > > >
> > > > I only mentioned "novelty" in my comment since I felt that was the main change, but I did indeed update some of the other scores (and the overall score).

---

### Official Review · Reviewer_Z6bA · 2021-11-06

**Correctness:** 4
**Technical Novelty And Significance:** 2
**Empirical Novelty And Significance:** 3
**Recommendation:** 8
**Confidence:** 3

**Main Review:**

The problem that they introduce and the solution are natural. I would say this part of the paper is interesting but unsurprising. Such generalizations are common and the tricks that this paper employes are not unusual. Still, it's a good contribution.

The stronger contribution in my view is the observation that this problem helps in the COVID vaccine design pipeline. I'm far from being an expert on this, so I'm not confident in my assessment, but it seems strong.





**Summary Of The Paper:**

This paper suggest the n-times coverage problem (a natural generalization of set-cover where items need to be covered n>1 times by the sets we choose) and motivates it with an application for the design of COVID vaccines. This application involves some ML pipeline, and this new combinatorial optimization problem helps with one of the steps.

The problem is shown to be not-submodular, and therefore the most standard methods don't quite work. Nonetheless, the authors present a greedy algorithm to get a reasonable approximation.

The paper evaluates the quality of their algorithm on artificial data and shows that it outperforms the more standard greedy method. Then they show the impact of this improvement for vaccine design.

**Summary Of The Review:**

This is a nice paper with good contribution for the (ML based) pipeline for COVID vaccine design.

---

### Decision · Program_Chairs · 2022-01-20

**Decision:**

Accept (Poster)

**Comment:**

The paper introduces the maximum n-times coverage, a new NP-hard (and non-submodular) optimization problem. It is shown that the problem can naturally arise in ML-based vaccine design, and two heuristics are given to solve the problem. The results are used to produce a pan-strain COVID vaccine.

The reviewers and I think that this is an interesting paper with a compelling application. There were some concerns about theoretical novelty and biological accuracy but these were addressed during the author response period. Given this, I am delighted to recommend acceptance. Please incorporate the feedback in the reviews in the final version of the paper.